# Recurrence-Free Survival and Disease-Specific Survival in Patients with Pancreatic Neuroendocrine Neoplasms: A Single-Center Retrospective Study of 413 Patients

**DOI:** 10.3390/cancers16010100

**Published:** 2023-12-24

**Authors:** Stine Møller, Seppo W. Langer, Cecilie Slott, Jesper Krogh, Carsten Palnæs Hansen, Andreas Kjaer, Pernille Holmager, Marianne Klose, Rajendra Singh Garbyal, Ulrich Knigge, Mikkel Andreassen

**Affiliations:** 1ENETS Center of Excellence, Copenhagen University Hospital—Rigshospitalet, 2100 Copenhagen, Denmark; stinemoeller@yahoo.com (S.M.); seppo.langer@regionh.dk (S.W.L.); cecilie_slott5@hotmail.com (C.S.); jesper.krogh@dadlnet.dk (J.K.); carsten.palnaes.hansen@regionh.dk (C.P.H.); akjaer@sund.ku.dk (A.K.); pernille.holmager.01@regionh.dk (P.H.); marianne.christina.klose.01@regionh.dk (M.K.); rajendra.singh.garbyal@regionh.dk (R.S.G.); ulrich.peter.knigge@regionh.dk (U.K.); 2Department of Endocrinology and Metabolism 7562, Copenhagen University Hospital—Rigshospitalet, 2100 Copenhagen, Denmark; 3Department of Oncology, Copenhagen University Hospital—Rigshospitalet, 2100 Copenhagen, Denmark; 4Department of Clinical Medicine, University of Copenhagen, 1172 Copenhagen, Denmark; 5Department of Surgery and Transplantation, Copenhagen University Hospital—Rigshospitalet, 2100 Copenhagen, Denmark; 6Department of Clinical Physiology and Nuclear Medicine & Cluster for Molecular Imaging, Copenhagen University Hospital—Rigshospitalet, 2100 Copenhagen, Denmark; 7Department of Biomedical Sciences, University of Copenhagen, 1172 Copenhagen, Denmark; 8Department of Pathology, Copenhagen University Hospital—Rigshospitalet, 2100 Copenhagen, Denmark

**Keywords:** pancreatic neuroendocrine neoplasms, incidence, clinical presentation, prognosis

## Abstract

**Simple Summary:**

The survival and disease recurrence for pancreatic neuroendocrine tumors differs greatly. The aim of our retrospective study was to investigate prognostic factors influencing recurrence and survival of 413 patients. For outcome evaluation the cohort was split in three groups: those under surveillance, those who had surgery with the intention to cure, and those with unresectable disease. For small tumors (<2 cm) under surveillance only 1 of 59 required surgery. For those who had surgery 20% had recurrence within 5-years with tumor size, high Ki-67 index, and location in the pancreatic head as risk factors. Half of the patients with unresectable tumors survived longer than approximately 2 years with high age, high chromogranin A levels, and increased markers of tumor proliferation being linked to survival. In conclusion, the current results add valuable knowledge in terms of predicting outcomes for individuals with pancreatic neuroendocrine neoplasms.

**Abstract:**

Introduction: The prognosis and impact of different prognostic factors in pancreatic neuroendocrine neoplasms (pNEN) remain controversial. Aim: To investigate prognostic factors for recurrence-free survival and disease-specific survival in patients with pNEN, divided into three groups: patients undergoing surveillance (tumor size < 2 cm, group 1), patients followed after curative-intended surgery (group 2), and patients with unresectable disease or residual tumors after resection (group 3). Method: A single-center retrospective study including consecutive patients over a 20-year period. Multivariate Cox regression analyses were performed to identify risk factors. Results: 413 patients were included, with a mean (SD) age of 62 ± 14 years. In group 1 (n = 51), median (IQR) follow-up was 29 (21–34) months, and tumor size was 1.0 (0.8–1.4) cm. One progressed and had a tumor resection. In group 2 (n = 165), follow-up 59 (31–102) months, median tumor size 2 (1.2–3.4) cm, median Ki-67 index 5 (3–10)%, the 5-year recurrence rate was 21%. Tumor size (*p* < 0.001), Ki-67 index (*p* = 0.02), and location in the pancreatic head (*p* < 0.001) were independent risk factors. In group 3 (n = 197), follow-up 19 (6–46) months, median tumor size 4.2 (2.6–7.0) cm, Ki-67 index 17 (9–64)%, the median disease-specific survival was 22 (6–75) months—99 in NET G1; 54 in NET G2; 14 in NET G3; and 6 months in neuroendocrine carcinomas (NEC). Age (*p* = 0.029), plasma chromogranin A (*p* = 0.014), and proliferation, expressed by grade (*p* = 0.001) and Ki-67 index (*p* < 0.001), were risk factors. Conclusion: Growth in pNET < 2 cm requiring surgery was observed in 1/51. Tumor size, Ki-67 index, and location in the head were prognostic factors for disease recurrence, while age, plasma chromogranin A, and proliferation predicted mortality in patients with unresectable disease or residual tumors after resection.

## 1. Introduction

Gastroenteropancreatic neuroendocrine neoplasms (GEP-NENs) encompass a heterogeneous group of neoplasms sharing a common histopathological feature. They are categorized into two groups: well-differentiated neuroendocrine tumors (NET) and poorly differentiated neuroendocrine carcinomas (NEC). NETs are further classified based on their Ki-67 index into low-grade (NET G1, Ki-67 < 3%), medium-grade (NET G2, Ki-67 3-20%), and high-grade (NET G3, Ki-67 > 20%) tumors and neuroendocrine carcinomas (NEC) G3 with low differentiation [1]. The majority of pancreatic NENs (pNENs) are well-differentiated non-functioning tumors [2,3,4]. During the last 50 years, numerous studies have documented a significant increase in the incidence of pNEN [5,6,7]. A possible explanation for this rise includes an increased number of incidentally discovered tumors, greater awareness among pathologists, and a true increase in incidence [6].

Treatment decisions and prognosis primarily rely on tumor grade, size, and stage [8,9,10]. Watchful waiting has been deemed safe for older patients with low-grade localized pNET measuring ≤2 cm in size [8]. For all other tumors, regardless of stage and grade, surgery should always be considered due to the risk of the development of metastatic disease [11,12]. In cases of unresectable disease, various palliative medical options are available [8,12,13,14].

Patients with disseminated disease have a median life expectancy of around 5 years for low-grade tumors and 2 years for high-grade tumors [4,15,16,17]. However, many previous studies are non-European and were initiated several decades ago, failing to account for the new treatments and grading standards, including the introduction in 2016 of a novel tumor category: NET G3, i.e., well-differentiated tumors with Ki-67 > 20% [18,19]. Targeted therapies like everolimus and sunitinib were approved for pancreatic NET G1 and NET G2 approximately 10 years ago, while somatostatin receptor targeting Peptide Receptor Radionuclide Therapy (PRRT) has primarily been utilized in European countries for more than two decades [14,20].

We present data on the prognosis of patients with pNEN treated at the Copenhagen ENETS Center of Excellence, Rigshospitalet, during a 20-year period. The cohort was divided into three clinically relevant groups, and the primary outcomes were: (1) progression during surveillance of small unresectable tumors; (2) prognostic factors for recurrence-free survival (RFS) and disease-specific survival following curative-intended surgery; and (3) prognostic factors for disease-specific survival in patients with unresectable disease or residual tumors after resection.

## 2. Materials and Methods

The Copenhagen Neuroendocrine Tumor Center of Excellence serves approximately 2.7 million residents in the eastern part of the country.

At the time of diagnosis, baseline data were prospectively entered in a database. The recorded data included demographic information, clinical presentation (symptomatic or incidentally discovered tumor, evidence of hormone secretion, and germline mutations), primary tumor size (preferably based on pathology, if not available based on imaging), tumor location (divided into head vs. tail), disease stage (categorized as local disease (T1-T4N0M0), regional disease with intraabdominal lymph node metastases or disseminated disease with distant metastases including extra abdominal lymph node metastases), functioning vs. non-functioning, immunohistochemistry markers (Ki-67, chromogranin-A (CgA), synaptophysin) and somatic mutations through next generation sequencing (only NET G3 and NEC) (NGS) [21]. Follow-up data were obtained from medical records as part of this study protocol and included treatment, RFS after surgery, and disease-specific survival.

### 2.1. Patient Selection

Consecutive patients diagnosed with NEN arising in the pancreas or ampulla of Vater and followed from 1 January 2000 to 31 December 2020, were included in this study. The diagnosis was based on cytohistomorphology and immunostaining for synaptophysin and CgA. Tumors were graded according to the WHO grading system as NET G1-G3 or NEC. In cases where histological diagnosis was unavailable, the diagnosis was based on positive somatostatin receptor imaging (SRI), defined as higher uptake than physiological uptake in the liver (Krenning scale > 3). A diagnosis of NEC was based on cyto-histomorphology, Ki-67 proliferation’s index, immunohistochemistry, and mutational analysis [22].

Patients were followed until death or the end of the follow-up period on 31 December 2021. Due to a unique nationwide personal identification code in Denmark, no patients were lost to follow-up in the mortality analyses. Cases were excluded from this study if the histopathological examination revealed mixed neuroendocrine and non-neuroendocrine neoplasms (MiNEN).

### 2.2. Outcome

For outcome evaluation, the cohort was divided into 3 groups:

Group 1: Asymptomatic patients diagnosed with non-functioning localized pNET ≤ 2 cm who were assigned to watchful waiting with imaging every 6–12 months [8]. The outcome assessed in this group was progression, defined as an increase in tumor size leading to surgery, the discovery of metastatic disease, or disease-specific mortality. The cause of death was determined to be disease-specific if a comprehensive examination of the electronic patient records excluded other causes.

Group 2: Patients who underwent curative-intended surgery and had no detectable residual tumor tissue on the first postoperative imaging. The outcomes assessed in this group were RFS and disease-specific survival.

Group 3: Patients with unresectable disease, inoperable patients, or patients who underwent surgery, either palliative or curative-intent, but had residual disease on the 1st postoperative imaging. The outcome assessed in this group was disease-specific survival.

### 2.3. Statistics

Statistical analyses were performed with SPSS Statistics (IBM, version 28.0.0.0), and two-sided *p* values ≤ 0.05 were considered statistically significant. Descriptive statistics were used to summarize patient demographics and clinical characteristics. Unless otherwise specified, normally distributed variables are presented as mean (SD), while non-normally distributed variables are presented as median (inter quartile range (IQR)). Baseline continuous variables were compared using an unpaired *t*-test, while categorical data were compared using a χ^2^-test. To approximate a normal distribution, plasma CgA and Ki-67 indexes were log2 transformed.

Disease-specific survival and RFS were plotted using Kaplan–Meier curves for the entire cohort and stratified by each of the three outcome groups and by tumor grade. Differences across the different groups were assessed using the log-rank test.

Using univariate Cox proportional hazard regression analyses, the prognostic importance (hazard ratio, HR) of different potential prognostic factors, including gender, age at diagnosis, year of diagnosis, tumor grade, stage, incidentaloma, functioning tumor, location and size of the primary tumor, p-CgA, and Ki-67 index, were assessed. Subsequently, all covariates with *p* < 0.2 were included in a multivariate Cox hazard regression analysis with backward elimination (conditional method). The HR for p-CgA and Ki-67 was estimated per 2-fold increase in non-logarithmically transformed p-CgA and Ki-67, respectively.

## 3. Results

### 3.1. Baseline Characteristics

The entire cohort consisted of 413 patients with pNEN with a mean age of 62 ± 14 years (57% men). The median follow-up period was 35 (14-69) months. To analyze time trends, the cohort was divided into two 10-year periods: 2000–2009 and 2010–2020. More patients were diagnosed in the latter period (329 vs. 84). The mean age at diagnosis was significantly lower in the former period (57 ± 15 vs. 63 ± 14 years, *p* < 0.001). The baseline characteristics for the entire cohort and for the three different groups are presented in Table 1. Most tumors, 195/351 (55%), were classified as NET G2. At diagnosis, 193/412 (47%) had local disease, 48/412 (12%) had regional disease with abdominal lymph node involvement, and 171/412 (41%) had disseminated disease with distant metastases. Fifty-two of 413 (13%) tumors were classified as functioning, with insulinomas (n = 33) and gastrinomas (n = 10) as the most frequent hormone-secreting tumors. In total, 174/396 (44%) tumors were diagnosed incidentally (incidentaloma), 35/51 (69%) in group 1 (surveillance), 82/165 (50%) in group 2 (curative intended surgery), and 57/197 (29%) in group 3 (no curative intended surgery). Incidentalomas were defined as an unexpected finding on imaging not related to the original diagnostic inquiry. There was a lower fraction of incidentalomas in the former period (2000–2009) compared to the later period (2010–2020) ((14/69 (20%) vs. 160/327 (49%), *p* < 0.001). The location of the primary tumor was significantly associated with tumor staging, with localized tumors without metastases presenting more often in the tail as compared to the head (102/178 (57%) vs. 69/171 (40%) (*p* < 0.001). In total, 185 patients had surgery for primary tumors—13 were carried out as enucleation (all insulinomas < 2 cm), while all other surgeries were carried out as formal pancreatectomies with lymph node resection (total pancreatectomy, distal pancreatectomy, or Whipple’s procedure). Survival in the total cohort and stratified per grade is presented in Figure 1A,B.

### 3.2. Group 1—Surveillance

Fifty-one patients with a mean age of 60 ± 15 years were followed by watchful waiting for a median of 29 (21–43) months. The median tumor size at baseline was 1.0 (0.8–1.4) cm. Diagnosis was based on histopathology in 9/51 (18%) cases and SRI in the remaining cases (Krenning scale > 3). One patient refused surgery for a small localized insulinoma and was treated with SSA to control symptoms. No other patients in the surveillance group received medical treatment to control tumor growth. Only 1 patient required surgery due to an increase in tumor size from 1.3 to 2.0 cm (Ki-67 4%) 12 months after diagnosis. Post-surgery, he has been followed for 100 months without recurrence. There were no disease-specific deaths in this group.

### 3.3. Group 2—Curative Intended Surgery

The mean age was 58 ± 14 years, and patients were followed for a median of 59 (31–102, range 1–254) months. The median tumor size at baseline was 2 (1.4–3.6) cm for non-functional tumors and 1.5 (1.0–2.5) cm for the 29 functional tumors. Among the 17 patients with germline mutations, only one had a recurrence (lymph node metastasis in a patient with MEN1). In all patients, staging based on SRI was conducted pre-surgery. All patients underwent thorough evaluation and discussion at our tumor board before the ultimate decision to proceed with curative surgery. The presence of distant metastases was not deemed a contraindication for surgery.

Recurrence-Free Survival

Recurrence occurred in 35 of 165 patients (21%). RFS rates were 88% after 2 years, 81% after 5 years, and 69% after 10 years (Figure 1C,D). Stratified by stage, 15/124 (12%) of patients with local disease had recurrence compared to 15/32 (47%) with regional disease and 5/9 (56%) with disseminated disease. Stratified by grade, the recurrence rates were 5/29 (17%, NET G1), 20/109 (18%, NET G2), 3/8 (38%, NET G3), and 7/10 (70%, NEC). Two patients received neo-adjuvant chemotherapy before resection.

In the multivariant Cox regression analysis, with proliferation expressed as WHO grade, location in the pancreatic head (HR 4.6, 95% CI: 1.9–11.3, *p* < 0.001) and size of primary tumor (HR per 1 cm increase, 1.4 (95% CI: 1.2–1.5, *p* < 0.001) were identified as independent risk factors for recurrence, whereas grade or stage did not predict recurrence (Table 2). When grade was replaced with the Ki-67 index as a measure of proliferation, proliferation emerged as an independent risk factor (HR per 2-fold increase in Ki-67, 1.4; 95% CI: 1.1–1.9; *p* = 0.02) along with size and location (Table 2). Figure 2 illustrates the significant difference in RFS stratified by location in the pancreas. The latest recurrence was observed after 167 months with liver metastasis (primary tumor characteristics were Ki-67 1%, size 2.5 cm, regional disease with one lymph node metastasis).

Disease-Specific Survival

Among the 165 primarily curative-operated patients, 17 (10%) died from pNEN (Figure 1E). In the subsequent multivariant Cox regression analysis, with proliferation expressed as grade, only grade was significantly associated with disease-specific survival (*p* < 0.001). In pairwise comparison, the only significant difference was between NETG1 and NEC (for details, see Table 3). When grade was replaced with the Ki-67 index, proliferation remained an independent risk factor (HR per 2-fold increase in Ki-67: 2.4; 95% CI: 1.6–3.4, *p* < 0.001, for details see Table 3).

### 3.4. Group 3—Unresectable Disease or Residual Tumor after Resection

The mean age was 66 ± 13 years, and the median follow-up was 19 (6–46, range 0–219) months. The Ki-67 index was available in 185 of 197, with a median value of 17 (9–64%). A total of 163 patients received systemic treatment following diagnosis. There have been minor changes in treatment strategies over time, e.g., increased use of everolimus in NET G2 and increased focus on temozolamide and capecitabine as the 1st line in NET G3. However, throughout this study period, chemotherapy with streptozotocin/5-fluorouracil was used as 1st line therapy and PRRT as 2nd line therapy for most patients with NET G2 (for details, see Table 4. Out of the 20 primary surgical approaches in group 3, 5 were carried out as palliative surgery without a curative intent.

Disease-Specific Survival

Of 197 patients with unresectable disease or residual tumors after resection, 144 (73%) died of pNEN. The median disease-specific survival was 22 (6–75) months. The 2-year disease-specific survival rate was 48%, the 5-year disease-specific survival rate was 30%, and the 10-year disease-specific survival rate was 16% (Figure 1G,F). Independent risk factors for disease-specific survival were age (HR 1.0, *p* = 0.029), p-CgA (HR per 2-fold increase 1.1, 95% CI: 1.0–1.2, *p* = 0.014), and proliferation expressed as grade (*p* < 0.001). In pairwise comparison, there was a significant difference between NET G1 and NET G3 and between NET G1 and NEC, but there was no difference between NET G1 and NET G2 (HR 1.4 (95CI 0.6–3.0); for details, see Table 5 and Figure 1H). The median survival was 99 months in NET G1, 54 months in NET G2, 14 months in NET G3, and 6 months in NEC. When grade was replaced with Ki-67 index, similar results were obtained (HR per 2-fold increase in Ki-67 1.7; 95% CI: 1.5–2.0; *p* < 0.001) (for details, see Table 5 and Figure 1H). Stage was not an independent risk factor, regardless of whether proliferation was expressed as the Ki-67-index or grade.

The two longest-surviving patients with disseminated disease were followed for 18 years and were alive at the end of the follow-up period. Both patients had received PRRT and chemotherapy.

## 4. Discussion

This is the largest single-center study providing data on prognosis in patients with pNEN. We found that active surveillance in patients with localized small tumors (group 1) was associated with a very low risk of growth. After curative intended surgery (group 2), the 5-year risk of recurrence was approximately 20%, with tumor size, location in the head of the pancreas, and Ki-67 index being independent risk factors, while grade and stage did not predict recurrence. In patients with unresectable disease (group 3), the median disease-specific survival was 22 months, with proliferation, age, and p-CgA as independent risk factors.

The mean age at diagnosis in our cohort was 62 years, which corresponds to other studies [2,4,6,16]. We observed a time trend with increasing age at diagnosis, which is also consistent with previous publications and likely reflects a change in the patient population due to an increased incidence of incidentally found tumors in the elderly population [2,4,6,7,10,23]. In the earlier decade (2000–2009) compared to the later decade (2010–2020), the incidence of pNEN patients increased by a factor of 3, while the background population size remained unchanged [6]. We did not find any impact of time on overall disease-specific survival, in contrast to previous publications reporting better survival over time.

The data from the surveillance group (group 1) supports the notion that watchful waiting in elder individuals are a safe strategy for tumors <2 cm, as only one patient had tumor growth that required surgery and none died as a result of observation of the pNEN. The ENETS 2016 guidelines recommend surveillance with MRI or CT every 6–12 months and longer intervals if the tumor is <1 cm and stable. The question up for discussion is how long this follow-up period should be continued since studies [13,24,25] supporting the safety of observation have a short follow-up, including data from the present study.

Recurrence rates 5-years after curative-intended surgery (group 2) were approximately 20%. Despite the inclusion of patients with metastatic disease, the recurrence rate in our study remained comparable to the findings reported in a recently published metaanalysis, which primarily comprised patients with non-metastatic disease. Additionally, our results align with a recently published single-center study that also included patients with metastatic disease [16,26]. The selection of the right patient for intended curative surgery is debated, especially in cases of metastatic disease and high-grade tumors. To ensure a consistent and standardized approach, all patients were subject to review and discussion at our tumor board prior to the definitive decision to undergo surgery. However, we acknowledge that the decision-making process might have been influenced by the surgeon’s experience and local practice, which can vary. In our study, stage at diagnosis was not found to be an independent risk factor for recurrence after surgery, and some of our patients with high-grade tumors did not have recurrence and were long-term survivors. Therefore, our data support the idea that intended curative surgery should be considered in selected patients with metastatic disease and/or high-grade tumors.

As depicted in the Kaplan–Meier curve (Figure 1C), recurrence occurred without any apparent flattening of the curves for 10 years. Only one patient had a recurrence more than 10 years post-surgery. The location in the head of the pancreas was identified as a strong risk factor for recurrence. Only one out of 66 patients with tumors in the tail measuring <3 cm experienced a relapse, while 13 out of 43 patients with a tumor in the head measuring <3 cm had a recurrence. One previous study also reported a significant correlation between tumors in the head of the pancreas and positive lymph node involvement, which aligns with our results [11]. We acknowledge that our result on the relationship between location and prognosis is based on low numbers and needs to be confirmed in future studies. In 2022, Andreasi et al. [16] proposed an algorithm for tailored follow-up after curative surgery. In contrast to the information gained from our data, the authors suggested closer follow-up for tumor grade NET G2-G3 compared to NET G1, and for patients with primary nodal involvement. In our cohort, proliferation was only an independent risk factor when expressed as the Ki-67 index (and not by grade), and the presence of lymph node metastasis was not identified as an independent risk factor. A preceding study demonstrated that the extent of lymph node involvement provided prognostic insights; however, this aspect was not investigated in the current study [27]. Overall, studies examining risk markers for recurrence have yielded different results, and currently, there is no universally accepted algorithm for deciding on surgery for borderline-sized tumors or determining the optimal post-surgery follow-up program. The results from the present study point towards size, proliferation, and location in the head of the pancreas as important risk factors for recurrence. The prognostic importance of the Ki-67 index supports the idea that biopsy of borderline-sized tumors might provide valuable information when deciding between surgery and watchful waiting, although a specific Ki-67 cut-off has not yet been identified.

In patients with unresectable disease or residual tumor after resection (group 3), age, proliferation, and p-CgA were identified as independent risk factors consistent with results from previous studies [3,15]. The total 5-year disease-specific survival was 30% (63% for patients with NET G1, 36% for patients with NET G2, 11% for patients with NET G3, and <5% for patients with NEC), similar to what was reported in a recent large American registry-based study (5-year all-cause mortality of 72%, n = 3812) [5]. Approximately 15 out of the 197 patients in group 3 underwent curative-intent surgery. However, since they still had residual disease on their initial post-operative imaging, we made the decision to categorize them within group 3. Alternatively, these patients could have been included in group 2, and this might have had an impact on the prognosis in groups 2 and 3. Our patients received different standard treatments, as listed in Table 4. In contrast to previous studies, we did not observe a better prognosis over time in patients with unresectable disease [3,5,28]. A contributing factor could be a change in indication for surgery, with more patients with more advanced disease having surgery in the latter period. When excluding patients in the surveillance group, we observed that 35% of patients underwent intended curative surgery in the period from 2000 to 2009, whereas this percentage increased to 49% in the period from 2010 to 2020.

Disease-specific survival in patients with disseminated disease was the only outcome where p-CgA provided prognostic information, supporting its relevance as a disease marker for progressive disease, although its clinical utility in everyday clinical practice has been questioned due to its low sensitivity and specificity [29].

In general, our study, along with other studies [8,30,31,32], supports the importance of a grading system based on proliferation but also raises questions about the optimal cut-offs in relation to prognosis. The results from the present study align with the updated ENETS 2016 guidelines [8,22], which distinguish between NET G3 and NEC and demonstrate a substantially worse prognosis for NEC patients in all outcomes. By contrast, no significant differences were found between NET G1 and NET G2 for any outcomes in both univariate and multivariate analyses. Thus, an update to the grading system might be needed, and we are currently planning a separate publication that will specifically focus on the optimal prognostic Ki-67 index cut-offs. The forthcoming study will combine data from the current cohort with a large cohort of patients with small intestinal NET.

The main strengths of this study include a large cohort, information from individual patient files and not registers, and a long follow-up period [15]. Furthermore, this study reports disease-specific survival and incorporates the new 2019 WHO grading system [19]. The main limitations include the retrospective design, which carries the risk of missing data, especially in the earlier period, potentially introducing temporal bias. Moreover, it is important to note that the limited number of events for certain outcomes has led to wide confidence intervals and potentially overfitting of the statistical models. The surveillance guideline for pNET < 2 cm is new, resulting in a short follow-up for these patients, who, as another weakness, often lacked a histopathological diagnosis. The surveillance group encompassed certain patients with germline mutations, which could potentially have a more benign growth pattern compared to sporadic cases. Due to changes in clinical management over time for non-functional pNET < 2 cm, group 2 included a significant number of patients with tumors <2 cm, which would probably not, in current practice, have been recommended for resection. Finally, it must be acknowledged that imaging techniques have changed from SRI with gamma cameras to SRI with PET, e.g., ^68^Ga-DOTATATE or ^64^Cu-DOTATATE, which offer higher sensitivity and detection rates for metastases.

## 5. Conclusions

There is still a long way to go in terms of personalized and tailored management of patients with pNENs. Overall, the data supports proliferation as a crucial predictive marker. However, it raises concern regarding the separation between NETG1 and NETG2 using a Ki-67 cut-off of 3%, as it provided limited prognostic information. Furthermore, stage was not found to be an independent risk factor for recurrence after surgery or disease-specific survival in disseminated disease, reinforcing that active surgical and oncological treatment should always be considered. Finally, the significance of primary tumor size was confirmed in both patients with tumors <2 cm undergoing active surveillance and after intended curative surgery.

## Figures and Tables

**Figure 1 cancers-16-00100-f001:**
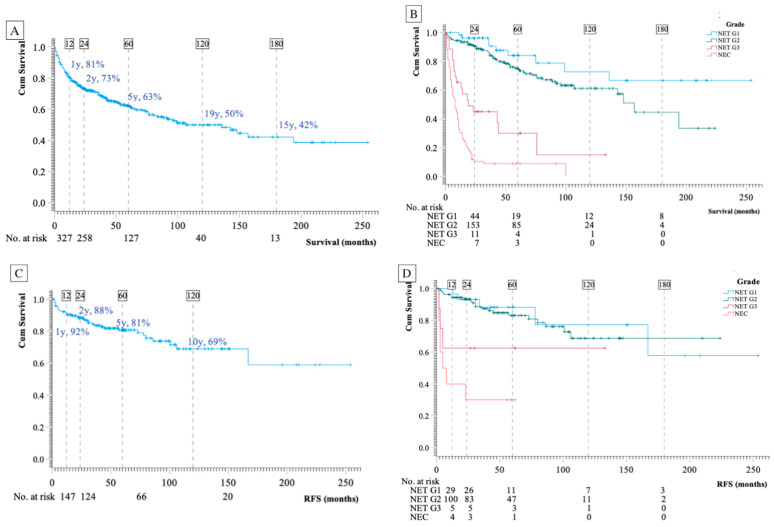
Kaplan–Meier curves present overall disease-specific survival and recurrence-free survival (RFS). Log Rank analyses for the stratified KM curves showed *p* < 0.001. Disease specific survival, total cohort (**A**,**B**). Recurrence free survival after curative surgery (**C**,**D**). Disease specific survival after curative surgery (**E**,**F**). Disease specific survival in disseminated disease (**G**,**H**).

**Figure 2 cancers-16-00100-f002:**
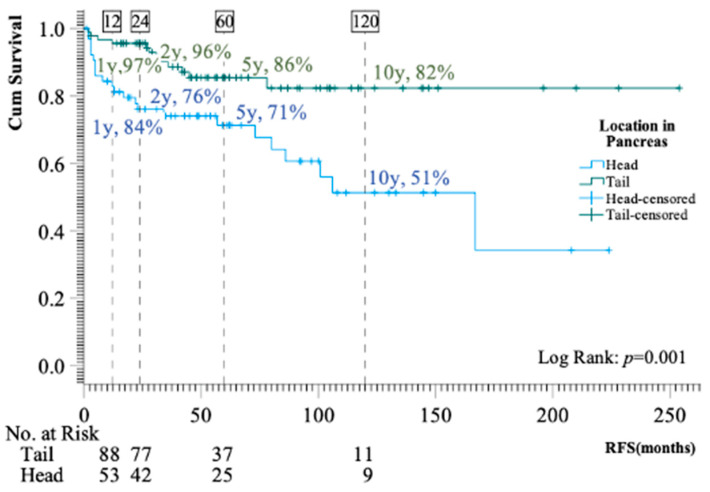
Kaplan–Meier curve presenting recurrence-free survival following curative surgery stratified by location in the pancreas (head vs. tail).

**Table 1 cancers-16-00100-t001:** Baseline characteristics.

		Overall	Surveillance (Group 1)	Curative Surgery (Group 2)	Unresectable Disease (Group 3)
		n = 413	n = 51	n = 165	n = 197
Age, years (SD)		62 ± 14	60 ± 15	58 ± 14	66 ± 13
Gender	Male	234 (57%)	27 (53%)	97 (59%)	110 (56%)
Year of diagnosis	2000–2009	84 (20)	1 (2)	29 (18)	54 (27)
	2010–2020	329 (80)	50 (98)	136 (82)	143 (73)
Incidentaloma		174 (44%)	35 (69%)	82 (50%)	57 (29%)
Heredity	MEN-1	25 (6%)	8 (16%)	12 (7%)	5 (3%)
	VHL	7 (2%)	4 (8%)	2 (1%)	1 (1%)
	NF1	3 (1%)	-	3 (2%)	
Functional tumor	Insulinoma	33 (8%)	1 (2%)	22 (13%)	10 (5%)
	Gastrinoma	10 (2%)	-	3 (2%)	7 (4%)
	Glucagonoma	2 (1%)	-	1 (1%)	1 (0.5%)
	VIP’oma	3 (1%)	-	1 (1%)	2 (1%)
	Somatostatinoma	4 (1%)	-	3 (2%)	1 (0.5%)
	Total functional tumors	52 (13%)	1 (2%)	29 (18%)	22 (11%)
Stage (n = 412)	Local	193 (47%)	51 (100%)	124 (75%)	18 (9%)
	Regional	48 (12%)		32 (19%)	16 (8%)
	Metastatic	171 (41%)		9 (6%)	162 (82%)
Primary tumor size (cm), median (IQR)		2.5 (1.3–4.8)	1.0 (0.8–1.4)	2.0 (1.2–3.4)	4.2 (2.6–7.0)
Liver metastases		160 (39%)		9 (6%)	151 (77%)
Bone metastases		27 (7%)			27 (14%)
Ki-67%, median (IQR) (n = 348)		10 (4–25)	2 (1–2)	5 (3–10)	17 (9–64)
CgA, pmol/L (n = 373)		122 (78–409)	123 (69–115)	91 (63–141)	423 (135–1385)
2019 WHO grade (n = 351)					
	NETG1	54 (13%)	8 (16%)	29 (18%)	17 (9%)
	NETG2	195 (47%)	2 (4%)	109 (66%)	84 (43%)
	NETG3	26 (6%)		8 (5%)	18 (9%)
	NEC	76 (18%)		10 (6%)	66 (34%
Location in pancreas (n = 350)	Head	172 (42%)	21 (41%)	65 (39%)	86 (44%)
	Tail	178 (43%)	18 (35%)	92 (56%)	68 (35%)
Primary surgery		185 (45%)		165 (100%)	20 (10%)

Baseline characteristics for the entire cohort were divided into groups. MEN-1 = multiple endocrine neoplasms 1, VHL = Von Hippel-Lindaus; NF = neurofibromitosis. CgA = chromogranin A. WHO = world health organization. VIP = vasointestinal peptide. Data on grade were not available in 21 of the 362 patients in group 2 or group 3, primarily due to a lack of Ki-67 staining in the early years.

**Table 2 cancers-16-00100-t002:** Recurrence following curative intended surgery (group 2).

Variable	Univariable Analysis		
	HR	95% CI	*p*
Sex, male vs. female (ref.)	1.0	0.5–2.0	0.97
Age at diagnosis, years	1.0	1.0–1.0	0.35
Location in head	3.0	1.5–6.0	0.002
Non-functioning (ref. functioning)	3.5	1.1–11.6	0.04
Not incidentaloma (ref. incidentaloma)	2.1	1.0–4.3	0.10
Primary tumor size, cm	1.2	1.1–1.3	<0.001
Stage (ref. localized)			<0.001
Regional	5.1	2.5–10.4	<0.001
Disseminated	12.2	4.3–34.9	<0.001
Grade (ref. NETG1)			<0.001
NETG2	1.2	0.5–3.3	0.71
NETG3	3.7	0.9–15.9	0.08
NEC	10.7	3.2–35.5	<0.001
Log2(CgA)	1.4	0.9–1.9	0.11
Log2(Ki-67)	1.8	1.4–2.2	<0.001
	**Multivariable analysis**		
With categorical Ki-67 (grade)	HR	95% CI	*p*
Primary tumor size, cm	1.4	1.2–1.5	<0.001
Location in head	4.6	1.9–11.3	<0.001
With continuous Ki-67			
Primary tumor size, cm	1.3	1.2–1.5	<0.001
Location in head	3.2	1.1–8.0	0.01
Log2(Ki-67)	1.4	1.1–1.9	0.02

Recurrence following curative intended surgery, showing harzard ratio (HR) and 95% confidence interval (95% CI) for prognostic variables in both uni- and multivariant analyses. Plasma Chromogranin (p-CgA) and Ki-67index were log (2) transformed, and HR is per 2-fold increase in non-logarithmical transformed CgA and Ki-67. The multivariable analyses were performed with Ki-67 as a categorical variable (grade) and as a continuous variable. NET = neuroendocrine tumor, and NEC = neuroendocrine carcinoma.

**Table 3 cancers-16-00100-t003:** Disease-specific survival following curative intended surgery (group 2).

		Univariable Analysis		
Variable		HR	95% CI	*p*
Sex, male vs. female (ref.)		1.1	0.4–2.8	0.90
Age at diagnosis, years		1.0	1.0–1.1	0.08
Location in head		3.3	1.2–9.4	0.03
Non-functioning (ref. functioning)		34.9	0.3–4114.0	0.14
Primary tumor size, cm		1.1	1.0–1.2	0.047
Stage (ref. localized)				0.006
	Regional	4.4	1.6–12.1	0.004
	Disseminated	6.5	1.3–32.1	0.022
Grade (ref. NETG1)				<0.001
	NETG2	3.2	0.4–28.6	0.29
	NETG3	10.9	1.0–122.3	0.05
	NEC	66.1	6.8–642.1	<0.001
Log2(CgA)		1.6	1.0–2.5	0.04
Log2(Ki-67)		2.3	1.7–3.2	<0.001
		**Multivariable analysis**		
With categorical Ki-67 (grade)		HR	95% CI	*p*
Grade (ref. NETG1)				<0.001
	NETG2	5.6	0.4–72.7	0.19
	NETG3	13.5	0.5–342.0	0.12
	NEC	169.2	9.3–3055.7	<0.001
With continuous Ki-67				
Age at diagnosis, years		1.0	0.99–1.1	0.06
Log2(Ki67)		2.4	1.6–3.4	<0.001

Mortality following curative intended surgery shows a hazard ratio (HR) and 95% confidence interval (95% CI) for prognostic variables in both uni- and multivariant analyses. Plasma Chromogranin (p-CgA) and Ki-67index were log (2) transformed, and HR is per 2-fold increase in non-logarithmical transformed CgA and Ki-67. The multivariable analyses were performed with Ki-67 as a categorical variable (grade) and as a continuous variable. NET = neuroendocrine tumor, and NEC = neuroendocrine carcinoma.

**Table 4 cancers-16-00100-t004:** Treatments.

Medical Treatments	n (%)
Somatostatin analogue	89 (21%)
Peptide-receptor radionuclide therapy	51 (12%)
Interferon	9 (2%)
Streptozotocin + 5-fluorouracil	77 (19%)
Everolimus	24 (6%)
Carboplatin + etoposide	68 (15%)
Topotecan	13 (3%)
Temozolomide	46 (11%)
Temozolomide + capecitabine	10 (2%)
Palliative radiation	26 (6%)
Others	12 (3%)

Total number of different treatments in the entire study population. Some patients received somatostatin analogs combined with another treatment.

**Table 5 cancers-16-00100-t005:** Disease-specific survival in patients with unresectable disease or residual tumors after resection (group 3).

		Univariable Analysis		
Variable		HR	95% CI	*p*
Sex, male vs. female (ref.)		1.0	0.7–1.3	0.80
Age at diagnosis, years		1.0	1.0–1.0	0.01
Decade of diagnosis (ref. 2000–2009)		1.3	0.9–2.0	0.13
Non-functioning (ref. functioning)		2.0	1.3–3.6	0.017
Not incidentaloma (ref. incidentaloma)		1.6	1.1–2.4	0.02
Stage (ref. localized)				0.01
	Regional	1.7	0.7–4.4	0.29
	Disseminated	3.0	1.5–5.9	0.002
Grade (ref. NETG1)				<0.001
	NETG2	1.5	0.7–3.2	0.29
	NETG3	4.6	1.9–20.0	<0.001
	NEC	10.0	4.6–21.8	<0.001
Log2(CgA)		1.1	1.0–1.2	0.02
Log2(Ki67)		1.7	1.5–2.0	<0.001
		**Multivariable analysis**		
With categorical Ki-67 (grade)		HR	95% CI	*p*
Age at diagnosis, years		1.0	1.0–1.0	0.029
Grade (ref. NETG1)				<0.001
	NETG2	1.4	0.6–3.0	0.457
	NETG3	4.7	1.8–12.2	0.001
	NEC	9.1	3.9–21.2	<0.001
Log2(CgA)		1.1	1.0–1.2	0.014
With continuous Ki-67				
Age at diagnosis, years		1.02	1.0–1.03	0.027
Log2(Ki67)		1.7	1.5–2.0	<0.001
Log2(CgA)		1.1	1.0–1.2	0.014

Mortality in patients with unresectable disease or residual tumor after resection shows a hazard ratio (HR) and 95% confidence interval (95% CI) for prognostic variables in both uni- and multivariant analyses. Plasma Chromogranin (p-CgA) and Ki-67 index were log (2) transformed, and HR is per 2-fold increase in non-logarithmical transformed CgA and KI-67. The multivariable analyses were performed with Ki-67 as a categorical variable (grade) and as a continuous variable. NET = neuroendocrine tumor, and NEC = neuroendocrine carcinoma.

## Data Availability

The data presented in this study are available on request from the corresponding author.

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
