# Peer review of "Recurrence-Free Survival and Disease-Specific Survival in Patients with Pancreatic Neuroendocrine Neoplasms: A Single-Center Retrospective Study of 413 Patients"

_cancers, 2023, doi:10.3390/cancers16010100_

Round 1

Reviewer 1 Report

Comments and Suggestions for Authors

Nice manuscript of over 400 patients with pancreatic neuroendocrine malignancies.

Overall important information, however not really new information.

The follow-up group ( group 1) also had some patients with MEN1 and VHL included. Why were these patients, with a genetic risk for pNEN included? Als one patient with an insulinoma was included in a wait and see protocol; unclear why! Table 1 states that Ki-67 was 2 % in group 1 but it is unclear from how many patients these data were available.

Group 2 is a cohort of curatively operated patients. How many patients were only diagnosed with a NEN after resection and could have been prevented to operate? Since the tumor size is so very small in this group, what was the reason for surgery?

the difference in survival after resection of a nen in the head versus in the tail is unclear; how many MEN patients for example had a "recurrence"or second primary? And since the patient groups are so small, ( 5 year only 37 versus 25 patients and 10 year only 11 versus 9 patients included) is this a relevant finding or is it based on insufficient / small data?

And in group 2 the patients with metastases are included;  Most studies exclude the genetic and the metastasized tumors form the risk analysis.

Multivariate analysis with 35 ( or 30 events if metastasized are excluded and even less events if genetic tumors are excluded) would result in max 2 items 

Why were excisiting risk predictors for recurrence after resection of pNEN not verified/used?

The statement that all PNEN should be evaluated for surgery should be adressed, since survival of the NEC and grade 3 was poor, even after resection.

some inconsistencies with tekst ( for example concerning missing data) and tables

unclear what the effect is of inclusion of insulinoma patients, since no recurrence is to be expected in those patients.

Comments on the Quality of English Language

inconsistencies  ( between text and data in tables, especially concerning missing data and patients) and language could be improved ( long sentences avoided)

Author Response

Reviewer 1

Nice manuscript of over 400 patients with pancreatic neuroendocrine malignancies.

Thank you for the appreciation of our study and the suggestions to improve the manuscript

Reviewer Overall important information, however not really new information.

The follow-up group ( group 1) also had some patients with MEN1 and VHL included. Why were these patients, with a genetic risk for pNEN included? Als one patient with an insulinoma was included in a wait and see protocol; unclear why! Table 1 states that Ki-67 was 2 % in group 1 but it is unclear from how many patients these data were available.

We agree that the biology of pNET in patients with germline mutations might be different from sporadic cases, and we have added this information under study limitations (page 14). However, to cover the total disease spectrum we prefer to have patients with germline mutations in the manuscript.

One insulinoma patients refused surgery of a small localized insulimona due risk of side effects. He was well-controlled on SSA. This information have been added to the manuscript  (page 7-8)

Diagnoses was based on histology in 9 of 51 patients in group 1. This information is already in the manuscript (page 7)

Group 2 is a cohort of curatively operated patients. How many patients were only diagnosed with a NEN after resection and could have been prevented to operate?

We agree that this could be interesting data, but unfortunately these data are not available in our data base

 Since the tumor size is so very small in this group, what was the reason for surgery?

Group 2 consist of non-functional and 29 functional tumors. The functional tumors were not resected based on tumor size and they were on average smaller than the non-functional tumors. We have added data om tumor size for non-functioning tumors (page 8) . Due to change in clinical management over time for non-functional pNET < 2 cm, group 2 included a significant number of  patients with tumors < 2cm, which would probably not in current practice have been recommended for resected. This information have been added to the manuscript (page 14).

The difference in survival after resection of a nen in the head versus in the tail is unclear; how many MEN patients for example had a "recurrence"or second primary?

Thank you for the possibility to clarify this matter. Recurrence was not due to a secondary primary in patients with germline mutations. Among the 17 patients with germline mutation only one had recurrence (LN metastasis). This information has been added to the manuscript (page 8)

And since the patient groups are so small, ( 5 year only 37 versus 25 patients and 10 year only 11 versus 9 patients included) is this a relevant finding or is it based on insufficient / small data?

We acknowledge that these results are based on low numbers, however the results are supported by an earlier study (ref 11) and we find that it is a potential interesting finding. The low numbers of patients have been added as a study limitation (page 12)

And in group 2 the patients with metastases are included;  Most studies exclude the genetic and the metastasized tumors form the risk analysis.

We agree that inclusion of patients in the different cohort can be debated and that it has been done differently in previous studies. We have made a comment in the discussion  (page 13)

Multivariate analysis with 35 ( or 30 events if metastasized are excluded and even less events if genetic tumors are excluded) would result in max 2 items 

We agree that the models might be overfitted - this information has been added to the manuscript (page 14)

Why were excisiting risk predictors for recurrence after resection of pNEN not verified/used?

In our opinion we have examined the most well established  risk factors such as size, proliferation, germline mutations etc . Further overfitting of the statistical models is obviously of concern if more variables should be included.

The statement that all PNEN should be evaluated for surgery should be adressed, since survival of the NEC and grade 3 was poor, even after resection.

Thank you – we have modified the statement (page 11-12)

some inconsistencies with text ( for example concerning missing data) and tables

Thank you – we have made changes in Table 1 and 3 (inconsistency between text and tables)

unclear what the effect is of inclusion of insulinoma patients, since no recurrence is to be expected in those patients.

We agree that the probability of recurrence in insulinoma is low, but it is not zero in our opinion justifying inclusion of these patients ( we have previous published data on malignancy in 80 insulinoma patients and found that) : DOI:10.1210/jc2019-01204

 Comments on the Quality of English Language

inconsistencies  ( between text and data in tables, especially concerning missing data and patients) and language could be improved ( long sentences avoided)

Thank you. We have tried to improve the language in the new text

Reviewer 2 Report

Comments and Suggestions for Authors

1. It would be interesting to know what criteria were applied to distinguish the curative intent group (Group 2) from the inoperable/unresectable group (Group 3). For example, although 77% of the inoperable/unresectable group were diagnosed with liver metastases (from Table 1), there are at least two American studies that show a survival benefit for primary site resection even in the presence of liver metastases. I would like to know that the authors considered clinical or experiential biases by surgeon in making determinations of who receives curative intent surgery versus a designation as inoperable. 

2. I would like to know the number of cases in Group 3 that were characterized as having curative-intent resection with residual disease. Also, I would like to know why these cases were included among the inoperable cases since it seems to me they would more appropriately be a subgroup of Group 2, curative intent surgery patients, for whom surgery was unsuccessful. It may be better to omit these cases or to treat them as a separate group in the analyses. 

3. I applaud the authors' use of up-to-date guidelines for classifying tumor grade using recent WHO criteria. I believe this information will help clinicians understand the different behaviors and clinical outcomes associated with the newer grading criteria.

4. Related to #3, I also appreciate that the authors have complete (please confirm) information on Ki-67 proliferation for patients. I find that this information is often absent in population-level data sets and limits conclusions that may be made about treatment outcomes when the influence of Ki-67 and other biomarkers on survival are unknown. 

5. It may be helpful for clinicians to have a clearer picture of the systemic therapies used in conjunction with surgery for Group 2, and if applicable, any systemic therapies received by patients in Group 3. 

Comments on the Quality of English Language

I note the use of the word "stabile" in 2nd sentence of the 2nd paragraph on page 6 "...if the tumor is <1cm and stabile." Is this a spelling error where the authors intended to write "stable"? 

Author Response

Reviewer 2

Dear reviewer – thank you for the suggestions to improve the manuscript.

  1. It would be interesting to know what criteria were applied to distinguish the curative intent group (Group 2) from the inoperable/unresectable group (Group 3). For example, although 77% of the inoperable/unresectable group were diagnosed with liver metastases (from Table 1), there are at least two American studies that show a survival benefit for primary site resection even in the presence of liver metastases. I would like to know that the authors considered clinical or experiential biases by surgeon in making determinations of who receives curative intent surgery versus a designation as inoperable.

This is a good point, and we have made appropriate changes under results (page 8) and in the discussion (page 11-12)

  1. I would like to know the number of cases in Group 3 that were characterized as having curative-intent resection with residual disease.

5 of 20 patients in group 3 who had surgery  were not  resected with a curative intend.  We agree that these data are important, and is now also included in the result section (page 9)

 Also, I would like to know why these cases were included among the inoperable cases since it seems to me they would more appropriately be a subgroup of Group 2, curative intent surgery patients, for whom surgery was unsuccessful. It may be better to omit these cases or to treat them as a separate group in the analyses. 

We agree that inclusion of patients in the different cohort can be debated and that it has been done differently in previous studies. From a patient point of view in relation to prognosis we find that it is more relevant to categorize these patients as having disseminated disease. But we agree that these patients could also be in group 2. If requested by the editor we can make changes. We have made a comment on this issue in the discussion (page 13)

  1. I applaud the authors' use of up-to-date guidelines for classifying tumor grade using recent WHO criteria. I believe this information will help clinicians understand the different behaviors and clinical outcomes associated with the newer grading criteria.

Thank you for the appreciation of our approach

  1. Related to #3, I also appreciate that the authors have complete (please confirm) information on Ki-67 proliferation for patients. I find that this information is often absent in population-level data sets and limits conclusions that may be made about treatment outcomes when the influence of Ki-67 and other biomarkers on survival are unknown. 

Among the 197 patiens in group 3 we had data on Ki-67 index in 185 patients – this data has been added (page 9 )

  1. It may be helpful for clinicians to have a clearer picture of the systemic therapies used in conjunction with surgery for Group 2, and if applicable, any systemic therapies received by patients in Group 3. 

Thank you for the suggestion to include more data on treatment. 2 patients in group 2 had neo-adjuvant treatment (page), this information has been added to the manuscript(page 8). We plan to  a separate publication on PFS of the most frequent used therapies in this cohort.

Comments on the Quality of English Language

I note the use of the word "stabile" in 2nd sentence of the 2nd paragraph on page 6 "...if the tumor is <1cm and stabile." Is this a spelling error where the authors intended to write "stable"?

Thank you – it is spelling error – has been changed
